# An Application of Safety Assessment for Radioactive Waste Repository: Non-Equilibrium Transport of Tritium, Selenium, and Cesium in Crushed Granite with Different Path Lengths

Chuan-Pin Lee [1,†], Dongyang Chen [1,†], Yanqin Hu [1], Yi-Lin Jan [2], Yunfeng Shi [1,3], Ziteng Wang [1], Enhui Wu [1], Neng-Chuan Tien [4,*], Yuzhen Sun [1] and Shih-Chin Tsai [4,*]

1   School of Nuclear Science and Engineering, East China University of Technology, Nanchang 330013, China; bennis6723@139.com (C.-P.L.); chen_dongyang2021@163.com (D.C.); hyqdrx@163.com (Y.H.); syf541006935@126.com (Y.S.); wzt2844@163.com (Z.W.); 201960370@ecut.edu.cn (E.W.); sunyuzhen1210@126.com (Y.S.)
2   Department of Civil Engineering, Chien Hsin University of Science and Technology, Taoyuan City 32097, Taiwan; yljan@uch.edu.tw
3   Department of Nuclear Environmental Science, China Institute for Radiation Protection (CIRP), Taiyuan 030006, China
4   Nuclear Science and Technology Development Center, National Tsing Hua University, Hsinchu 30013, Taiwan
*   Correspondence: nctien@mx.nthu.edu.tw (N.-C.T.); sctsai@mx.nthu.edu.tw (S.-C.T.); Tel.: +86-182-5824-4042 (N.-C.T.); +886-3-5715-131 (S.-C.T.)
†   These authors contributed equally to this work.

**Abstract:** Advection-dispersion experiments (ADE) were effectively designed for inadequate transport models through a calibration/validation process. HTO, selenium (Se), and cesium (Cs) transport in crushed granite were studied using a highly reliable, dynamic column device in order to obtain the retardation factors (R) and the dispersion coefficients (D) by fitting experimental breakthrough curves (BTCs) for various path lengths. In order to conduct a safety assessment (SA) of a deep geological repository for high-level radioactive waste, radionuclide transport in rock systems is necessary to clarify and establish a suitable model. A dynamic column with a radiotracer (HTO, Se(IV), and Cs) was applied to 2, 4, and 8 cm path lengths using a STANMOD simulation. The results showed similar results between the BTCs of Se and Cs by fitting a non-equilibrium sorption model due to the retardation effect. In fact, there was a relatively obvious sorption of Se and Cs in the BTCs obtained by fitting a retardation factor (R) value higher than 1. In addition, a two-region (physical) and a two-site (chemical) non-equilibrium model with either the lowest sum of squared residuals (SSQ) or the root mean square error (RMSE) were applied to determine the Se and Cs sorption mechanisms on granite.

**Keywords:** non-equilibrium; selenium; cesium; retardation factor; dispersion coefficient; breakthrough curves; STANMOD simulation

## 1. Introduction

In China, the field of nuclear technology, including such advances as Nuclear Power Plants (NPPs), is advancing rapidly and will likely revolutionize human life at the global level. Nuclear energy is now an important clean energy with zero carbon dioxide emissions [1,2]. In China, there are 54 nuclear power units in operation or under construction, with an installed capacity of 53 GW, accounting for about 5% of the total installed capacity, ranking third in the world [2,3]. Studies have been conducted on the development of nuclear power in China that provide a good overview of the required uranium fuel and thus nuclear waste to be produced [4,5].

Recently, applications of nuclear technology in science and technology as well as in healthcare have also undergone a dramatic increase. Nuclear technology has advantages

including generating a large amount of electricity and providing medical diagnoses while keeping wastes in a solid form within the plant. Of these wastes, the most serious in terms of safe management is the spent nuclear fuel, which contains some 99% of the total radioactivity. Bringing about a permanent disposal route for used fuel or high-level waste (HLW) is a political and scientific challenge that has yet to be fully met in any nuclear-powered country. In the case of HLW disposal, it is possible that people will be exposed to radiation. In fact, a HLW repository is designed with a multi-barrier system to achieve the purpose of long-term isolation of waste [5].

A report titled National Nuclear Long- and Medium-Term Development Planning (2005–2020) stated that the reactors completed in 2020, together with the 18 reactors under construction at that time, will eventually produce 82,630 tHM of spent fuel. As of 31 December 2010, the spent fuel stock was 2690 t in one nuclear power plant [3,4], in which the glass solidification body of high-level radioactive nuclear waste (HLW) was 673 m$^3$ and the cement solidified body of medium-level radioactive nuclear waste was 26,900 m$^3$ [5]. However, the large amounts of HLW have become a major safety and environmental issue [6]. Deep geological disposal of HLW is considered to be the safest and most feasible method. Geological disposal of HLW in the form of a "multi-barrier system" design comprising both man-made and natural barriers has been adopted for disposal repositories [7].

HLW such as spent fuel and vitrified glass are stored in waste cans and wrapped with buffer/backfill materials, and the outermost layer of nuclear wastes is the surrounding rock (including granite or clay rock, among others). To safely assess the most important "multi-barrier" repository, migration of radionuclides (RNs) in groundwater environments, the unique properties of radionuclides exhibit adequate physical and chemical behavior, including radiation activation, adsorption capacity, and surface behavior (interaction of clay or rock barriers) for migration in deep geological environments [8,9].

Safety assessments (SAs) of the final disposal of high-level radioactive waste (HLW) in geology require research on migration of important radionuclides in the surrounding environment. Generally, several important parameters for migration of radionuclides are applied in the transport of geological media for SAs, including the hydrodynamic dispersion coefficients ($D$), the diffusion coefficients ($D_e$), the retardation factor ($R$), and the distribution coefficients ($K_d$). The advection and dispersion behavior of radionuclides in granite fractures are of importance to maintaining the safety of geological disposal. Radionuclides not only can transport into the pore network of fractures or rock matrices, but also can sorb onto rock particles, in turn delaying their transport through the geosphere. The transport of radionuclides in the cracks in crystalline rocks can be described using a dual-porosity model [10]. These studies demonstrate that RN migration is dominated by various micro-processes, including anion exclusion, rock interactions, and diffusion in the rock matrix [11]. In view of the complexity of interactions between RNs and the mediums, knowledge about the migration behavior of RNs in granite fractures is far from complete. Cs is a highly yielded fission product containing up to 6.5% $^{135}$Cs ($t_{1/2} = 2.0 \times 10^6$ years), and $^{79}$Se is also a fission product with a long half-life of $3.56 \times 10^5$ years. Both Se and Cs belong to anion and cation respectively, as well as having high water solubility in groundwater, and Se and Cs can also be easily incorporated into terrestrial and aquatic organisms, such as occurred after the Fukushima accident, where radioactive wastewater leaked into the Pacific Ocean [12,13].

Granite is considered a potential host rock for disposing of spent nuclear fuel in China and several other countries. The fracturing of granite is the dominant pathway through which radionuclides transport from repositories to the biosphere. In this study, column experiments conducted in laboratories are used to study the retardation factors ($R$). By packing crashed granite samples into a column and conducting an advection-dispersion experiment at a low flow rate, it is possible to simulate the transport of radionuclides and thus determine the $R$. In order to develop a reliable model and obtain various parameters for long-term safety assessments, it is necessary to determine and quantify such advection and dispersion processes related to fractures or rock matrices. The free pore spaces between

these granitic grains and intragranular secondary pores form water-filled, heterogeneous networks with varying geometric factors, such as constrictivity and tortuosity [14,15]. Non-equilibrium transports are widely applied and studied in the literature [16,17] and can be classified into two models: the so-called physical non-equilibrium model and the chemical non-equilibrium model. The physical non-equilibrium model assumes that the porous media consists of mobile and immobile zones. Solute exchanges between these two zones follow a diffusion-controlled process. Sorption in these two zones is assumed to be instantaneous. The chemical non-equilibrium model assumes that sorption occurs on two different sites. In the first site, the sorption is instantaneous, while in the second site, the sorption is kinetic.

The object of this work is to investigate the sorption behavior of Se(IV) and Cs in a rough single fracture under different path lengths. The probable sorption mechanism of Se(IV) and Cs in a single fracture or in micro-pores is discussed in this paper. HTO, a nonreactive radiotracer, is applied and simulated prior to Se(IV) and Cs experiments, and the breakthrough curves (BTCs) of HTO, Se(IV), and Cs for different path lengths are investigated in order to obtain the dispersion coefficients (D), retardation factor (R), and dispersivity ($\alpha$) with crushed granite for a compacted column. In addition, the advection and dispersion behavior of Se(IV) and Cs in crushed granite is investigated and simulated to determine different reactive transport models. Finally, the retardation mechanisms of Se(IV) and Cs on granite rock could be generally demonstrated with a comparative analysis of the reactive transport models and the transport parameters. The results provide the important techniques and experimental data for SAs of HLW disposal in the near future.

## 2. Materials and Methods

### 2.1. Materials and Mineral Composition: Micro-Polar Microscopy and Micro-X-ray Computed Tomography

A mineral analysis using X-ray diffraction (XRD) of granite in rock was conducted in a previous study [18], where the main minerals were shown to be quartz, feldspar, biotite, and muscovite in the XRD spectra and micro-polar microscopy images [18–20].

From the X-ray attenuation, it is expected that signals' attenuation is an important parameter, and it is calculated and averaged from three-dimensional images over the volume of each part for scanning pixels. On the other hand, the micro-pore or fracture images between the mineral grains could be easily found by adjusting the X-ray signal noise, intensity, or resolution. In this work, three-dimensional images of an intact granite sample were tested and produced using a micro-X-ray computed tomography (SkyScan 1076, Bruker microCT, Kontich, Belgium) table top scanner, which is a non-destructive method used to detect differences in the X-ray attenuation between different micro-pores and mineral components of an intact granite rock in comparison to XRD spectra and micro-polar microscopy images.

Before they were used in the advective-dispersion experiments, all the rock samples were pre-treated by cutting, crushing, and passing through 20- and 50-mesh sieves (<1 mm), after which they were washed several times for 15–30 min with deionized water (DIW).

### 2.2. Theory: Advective-Dispersion Equation (ADE)

Solute transport and chemical reaction are primary components to affect the radionuclide activity in porous media. Three solute transport mechanisms describe the motion of mass through fluid in porous structure media: advection, diffusion, and dispersion. Chemical reactions are a series of complex activities that take place between radionuclides and media, including acid–base reactions, oxidation–reduction reactions, dissolution of precipitates, surface complex reactions, and ion exchange. Therefore, the study of transport behavior and chemical reaction in porous media requires complete information about the internal structures of the porous media, which at this time is not possible to obtain.

Considering the solute transport of a conservative tracer in a flow field of porous media without chemical reactions, advection and hydrodynamic dispersion are the two

mechanisms for the movement of solute particles. When the axis of the modeling domain aligns with the direction of the velocity vector, the dispersion coefficient is based on the dispersivity and molecular diffusion occurring in the principal direction only, as shown in [21]. By taking a two-dimensional case as an example, dispersion coefficients can be written for longitudinal and lateral dispersion as:

$$D_x = \alpha_L \, (u_x / |u|) + \alpha_T \, (u_y / |u|) + D_m \tag{1}$$

$$D_y = \alpha_L \, (u_y / |u|) + \alpha_T \, (u / |u|) + D_m \tag{2}$$

respectively, where $D$ is the hydrodynamic dispersion, $\alpha_i$ is the dispersivity, and $u_i$ is the component of pore velocity $u$ in the $i$ direction. $D_m$ is the molecular diffusion coefficient. In a one-dimensional isotropic medium, the dispersion coefficient can be further simplified as:

$$D = \alpha u + D_m. \tag{3}$$

The frequently used one-dimensional advective-dispersive equation for solute transport in the porous media can be expressed as:

$$\frac{\partial C}{\partial t} = D \frac{\partial^2 C}{\partial x^2} - u \frac{\partial C}{\partial x} \tag{4}$$

where $C$ = solute concentration in the liquid ($M/L^3$), $D$ = dispersion coefficient ($L^2/T$), $t$ = time (T), $u$ = pore water velocity (L/T), and x = distance (L). If the solute is subjected to linear equilibrium sorption, then Equation (4) can be written as:

$$\frac{\partial C}{\partial t} = \frac{D}{R} \frac{\partial^2 C}{\partial x^2} - \frac{u}{R} \frac{\partial C}{\partial x} \tag{5}$$

The retardation factor, $R$, can be given by:

$$R = 1 + [(\rho_b * K_d)/\theta] \tag{6}$$

where $\rho_b$ = bulk density ($M/L^3$), $K_d$ = distribution coefficient ($L^3/M$), and $\theta$ = porosity (-). The initial condition for the equation applied in this study is:

$$C(x,0) = 0 \tag{7}$$

and the boundary conditions for the upper and the lower boundary are written respectively as:

$$C(0,t) = C_0 \tag{8}$$

$$\frac{\partial C(\infty, t)}{\partial x} = 0 \tag{9}$$

An alternative lower boundary condition could also be applied as:

$$\frac{\partial C(L, t)}{\partial x} = 0 \tag{10}$$

where $L$ is the column length and $C_0$ is the inlet concentration. van Genuchten et al. [17,22–24] have recommended using Equation (9) as the lower boundary condition for a column experiment since the derived analytical solution is simpler but still accurate in most situations. Applying dimensionless procedures, the dimensionless parameters were defined as:

$$T = ut/L; \; Z = x/L; \; P = uL/D; \; C_1 = C/C_0$$

Introducing these parameters into Equation (5) gives:

$$R\frac{\partial C_1}{\partial T} = \frac{1}{P}\frac{\partial^2 C_1}{\partial Z^2} - \frac{u}{R}\frac{\partial C_1}{\partial Z} \tag{11}$$

The exit concentration, $C_e$, is defined as:

$$C_e(T) = C_1(1, T) = \frac{1}{2}erfc\left[\left(\frac{P}{4RT}\right)^{\frac{1}{2}}(R - T)\right] + \frac{1}{2}\exp(P)erfc\left[\left(\frac{P}{4RT}\right)^{\frac{1}{2}}(R + T)\right] \tag{12}$$

Equation (12) can only be applied to linear equilibrium adsorption cases. For a non-equilibrium adsorption model, a similar but more complicated mathematical procedure is needed. Two-region (physical) and two-site (chemical) models were frequently applied for the possible non-equilibrium adsorption situations.

### 2.3. Methods: A Dynamic Column System

The homogeneous columns were filled with granite samples, which were sampled from a small island near Kinmen, Fujian Province, China. For the porous media (i.e., crushed granite powders) used in this study, the hydraulic conductivity was found to be around $10^{-6}$ cm s$^{-1}$ [14]. The bulk density ($\rho$) and the total porosity ($\theta$) of the granite columns are shown in Table 1. The ADE installation in the experiments comprised primarily a multi-channel water-supply peristaltic pump (MASTERFLEX L/S, Cole-Parmer Instrument Co., Chicago, IL, USA), PP (poly-propane) columns (modified type, Hsinchu ZeGi Industrial Co., Ltd., Hsinchu, Taiwan), the solution tanks (radiotracer and synthetic groundwater (GW)), and an auto-fraction collector, as shown in Figure 1. In addition, saturated granite columns with fixed diameters of 5 cm and various path lengths of 2, 4, and 8 cm were simulated at a fixed $2.0 \pm 0.2$ mL/min flow rate for one-dimensional solute transport in homogeneous porous media. In fact, the three-to-two switch valves were open only to reservoir No. 2 (GW) to ensure water saturation in each pore space in the compacted granite before the ADE experiments. The packed column was slowly eluted to approximately a 5–10 pore volume (200–500 mL) with GW at a flow rate of $2.0 \pm 0.2$ mL/min. During the water saturation process, 10 mL of effluent from each column was sampled every 30 min. The concentrations of Na, Mg, Ca, and K in the effluent were measured using inductively coupled plasma–optical emission spectrometry (ICP–OES, DV 7000, PerkinElmer, Waltham, MA, USA). The variations in the concentrations were found to be within 5% of the corresponding liquid phase concentrations, and it was determined that water saturation had been reached.

**Table 1.** The ADE experimental conditions in this study.

| Item | No. 1 | No. 2 | No. 3 |
|---|---|---|---|
| Length (cm) | 2 | 4 | 8 |
| Pore volume (mL) | 18.06 | 36.11 | 72.22 |
| * Bulk density ($\rho_b$, g/cm$^3$) | | 1.45 | |
| * Porosity ($\theta$) | | 0.46 | |
| * Particle size (mm) | | <1 | |
| * Flow rate (mL/min) | | $2.0 \pm 0.2$ | |
| * Radionuclide: | | HTO: 50 Bq/mL (V0 $\fallingdotseq$ 5000 mL) | |
| HTO A0 (dpm/mL) | | Se(IV): $1 \times 10^{-4}$ M | |
| Se and Cs Concentration $C_0$ (M) | | Cs: $1 \times 10^{-4}$ M | |

* No. 1 and No. 3 have the same value as No. 2.

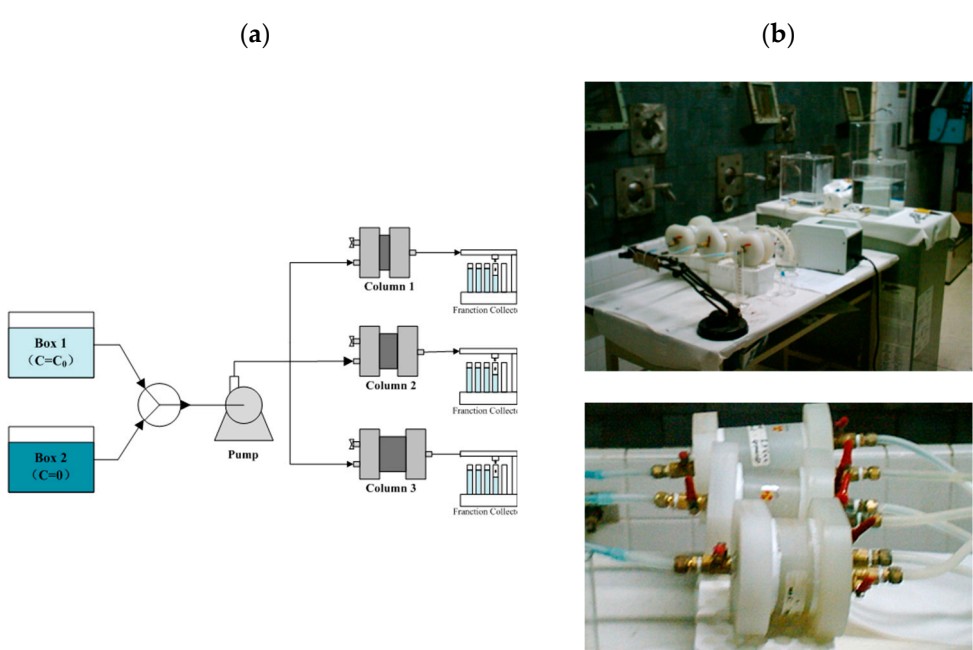

**Figure 1.** A reliable, precise ADE device with a PP (poly-propane) column apparatus used in this study. (**a**) schematic apparatus; (**b**) experimental set-up.

### 2.4. Experimental Set-Up for a Nonreactive Radiotracer (HTO) and a Reactive Radiotracer (Se and Cs)

Firstly, tritium ($^3$H, β-decay, and a half-life of approximately 12.3 years) served as the nonreactive radiotracer, and tritiated water (HTO) was used as a conservative or nonreactive tracer due to the fact that it is representative of all chemical elements that do not exhibit retention in the solid phase. The three sets of granite columns were prepared at the same time to observe the HTO, Se(IV), and Cs transport characteristics. Generally, the effluent was collected and measured for the purpose of plotting BTCs of different lengths from a series of up-flooding (sorption) processes [15,20,25]. The HTO samples were measured using a liquid scintillation analyzer (LSA, Perkin Elmer Tri-Carb 3170 TR/SL) with a counting efficiency of 51%, and 10 mL of the sample was combined with 10 mL of a scintillation cocktail (Packard LLT USA) in a 20 mL polyethylene counting vial. All chemicals used in the experiments were of analytical purity, and deionized water (DIW) was used throughout the experiments. After finishing the HTO experiments, both reactive radiotracers, Se(IV) and Cs, were spiked with selenium dioxide ($SeO_2$, Sigma Aldrich, Germany) and cesium chloride (CsCl, Merck, Germany) as stable isotope tracers, respectively. Both Se(IV) and Cs followed the same experimental process of up-flooding (sorption) processes in HTO. Three mL of the solution was reserved to perform the concentration analysis for Se(IV), using ICP–OES, and for Cs by the flame atomic absorption spectrometer (FAAS, iCE3000, Thermo, Germany).

### 2.5. Transport Simulation: HTO, Se, and Cs

The study of radionuclide transport behavior and chemical reaction in porous media requires complete information about the internal structures of the porous media, which, at the present time, cannot be obtained. Three mechanisms of solute transport describe the motion of mass through fluid in media with a porous structure: advection, diffusion, and dispersion. Chemical reactions are a series of complex equilibrium and non-equilibrium reactions between radionuclides and media, including acid–base reactions, oxidation–reduction reactions, precipitate dissolution, surface complex reactions, and ion exchange. Therefore, numerical simulations can be carried out using STANMOD (STudio of ANAlytical MODels), which is a package of programs used for predicting solute transport in saturated porous media using the analytical solutions for the classical advection-dispersion

equation [26]. Among those programs, CFITIM and CXTFIT can handle either the chemical or physical equilibrium and non-equilibrium transport issues. In this work, the equilibrium and the non-equilibrium model, so-called physical non-equilibrium simulations, were applied to simulate the experimental results of the radionuclide sorption kinetic mechanism for a solid/liquid system. Equation (12) can only be applied to linear equilibrium adsorption cases. For the non-equilibrium adsorption model, a similar but more complicated mathematical procedure is needed. Two-region (physical) and two-site (chemical) models were frequently applied for the possible non-equilibrium adsorption situations. The general dimensionless governing equations for the two non-equilibrium models are identical and are given as:

$$\beta R \frac{\partial C_1}{\partial T} + (1 - \beta) R \frac{\partial C_2}{\partial T} = \frac{1}{P} \frac{\partial^2 C_1}{\partial Z^2} - \frac{\partial C_1}{\partial Z} \tag{13}$$

$$(1 - \beta) R \frac{\partial C_2}{\partial T} = \omega(C_1 - C_2) \tag{14}$$

where (symbols with an identical definition as in the equilibrium model are not repeated here), for a two-region model, the dimensionless variables are given as:

$C_1 = C_m / C_0, C_2 = C_{im} / C_0,$

$C_m$ = solute concentration in the mobile liquid phase (M/L$^3$), $C_{im}$ = solute concentration in the immobile liquid phase (M/L$^3$),

$\beta = (\theta_m + f_{\rho_b} \times K_d) / (\theta + \rho b \times K_d),$

$\theta = porosity$, $\theta_m$ = porosity fraction of the mobile zone, $f$ = sorbed mass fraction assigned to mobile zone,

$\omega = \alpha L / \theta_m \times v_m,$

$\alpha = $ mass transfer coefficient (1/T),

$v_m = $ pore water velocity in the mobile zone (L/T).

For a two-site model, the dimensionless variables are given as:

$\beta = (\theta + F_{\rho_b} \times K_d) / (\theta + \rho_b \times K_d),$

$F = $ fraction of sites occupied by linear equilibrium adsorption,

$\omega = [\alpha(1 - \beta) RL] / v,$

$C_1 = C / C_0, C_2 = S_2 / [(1 - F) K_d C_0],$

$S2 = $ concentration subjected to kinetic non $-$ equilibrium adsorption (M/M).

The analytical solution of Equations (7)–(9), (13) and (14) is:

$$C_e = G(T) \exp \left( -\frac{\omega T}{\beta R} \right) + \frac{\omega}{R} \int_0^T G(\tau) H(T, \tau) d\tau \tag{15}$$

where

$$H(T, \tau) = \exp(-a - b) \big[ \frac{I_0(\xi)}{\beta} + \frac{\xi I_1(\xi)}{2b(1 - \beta)} \big] \tag{16}$$

$a = \omega \tau / \beta R,$

$b = (\omega(T - \tau)) / ((1 - \beta) R),$

$\xi = 2 (ab)^{(1/2)},$

$I_0, I_1$ = modified Bessel functions of the first kind.

$$G(\tau) = \tfrac{1}{2} erfc[(\tfrac{P}{4\beta R \tau})^{\frac{1}{2}} (\beta R - \tau)] +$$
$$\tfrac{1}{2} exp(P) erfc[(\tfrac{P}{4\beta R \tau})^{\frac{1}{2}} (\beta R - \tau)] \tag{17}$$

### 2.5.1. Two-Region Transport Simulation: HTO, Se, and Cs

In fact, the early breakthrough behavior of HTO BTCs indicated that HTO was subjected to non-equilibrium transport. Since HTO is nonreactive, this non-equilibrium transport is most likely induced by physical factors. That is, immobile zones probably existed in the experimental columns. We thus presumed that the fraction of mobile zones in each

column could be identified through analyzing HTO BTCs using a physical non-equilibrium model. The estimated parameters could then be used to calculate the pore water velocity, the dispersion coefficient, and the water content of the mobile region for each column.

2.5.2. Two-Site Simulation for Transport: HTO, Se, and Cs

Mathematically, the governing equations with dimensionless parameters for the physical and chemical non-equilibrium model are identical. However, the definitions of the dimensionless parameters are somewhat different when used to calculate the dimension variables for the two models. It thus might be interesting to determine how the transport variables would change if the chemical non-equilibrium model were to be implemented. The chemical non-equilibrium model is not appropriate to analyze HTO since HTO is nonreactive.

## 3. Results

### 3.1. Minerals and Micro-Structure Obtained Using Micro-X-ray Computed Tomography

Based on reports and literature from several National geological underground labs (Swiss Grimsel Labs, Sweden Äspö Labs, and Finland Olkiluoto Labs), the micro-CT resolution was set at 9 μm to determine and understand the micro-pore and mineral composition of the samples [11,27,28].

According to the micro-CT photos, there were coarse grain particles in the granite, as shown in Figure 2, because these rocks develop when volcanic (extrusive) igneous rocks cool slowly. In addition, fine micro-fractures and pores within intact granite were easily observed using contrasting (black/grey/white) CT images.

(**a**)　　　　　　　　　　(**b**)　　　　　　　　　　(**c**)

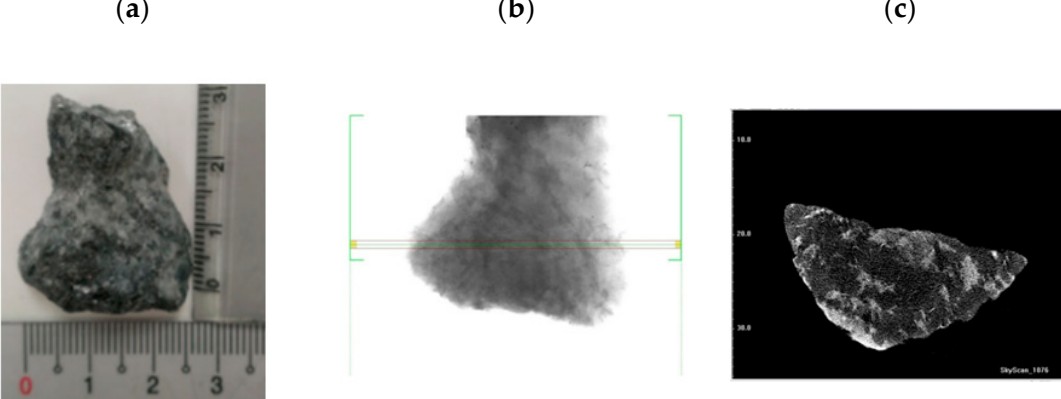

**Figure 2.** Micro-CT images of granite. (**a**) Intact granite sample, (**b**) horizontal scanning images in CT, and (**c**) vertical images in a tested granite sample.

### 3.2. Experimental Results for a Nonreactive Radiotracer (HTO) and a Reactive Radiotracer (Se and Cs)

A nonreactive radiotracer (HTO), prior to using Se(IV) and Cs, was applied to characterize the major physical transport processes in the proposed dynamic column system. Figure 3a shows that the HTO breakthrough curves (BTCs) at different lengths reached 1 ($C/C_0 \fallingdotseq 1$) from a series of up-flooding (sorption) processes. The results indicated that there were no dead-end or dead pores to block the HTO pathway, and water was saturated in each pore space in the compacted granite powder. However, Figure 3 shows that there was an obvious difference in the BTCs of Se(IV) and Cs ($C/C_0 < 1$) because of a higher retardation effect (R >1) than that for HTO, where the retention effect may have only been caused by sorption reactions. Among the BTCs, the BTCs of Cs at 8 cm revealed the lowest up-flooding process and an obvious retention effect, based on the experimental data.

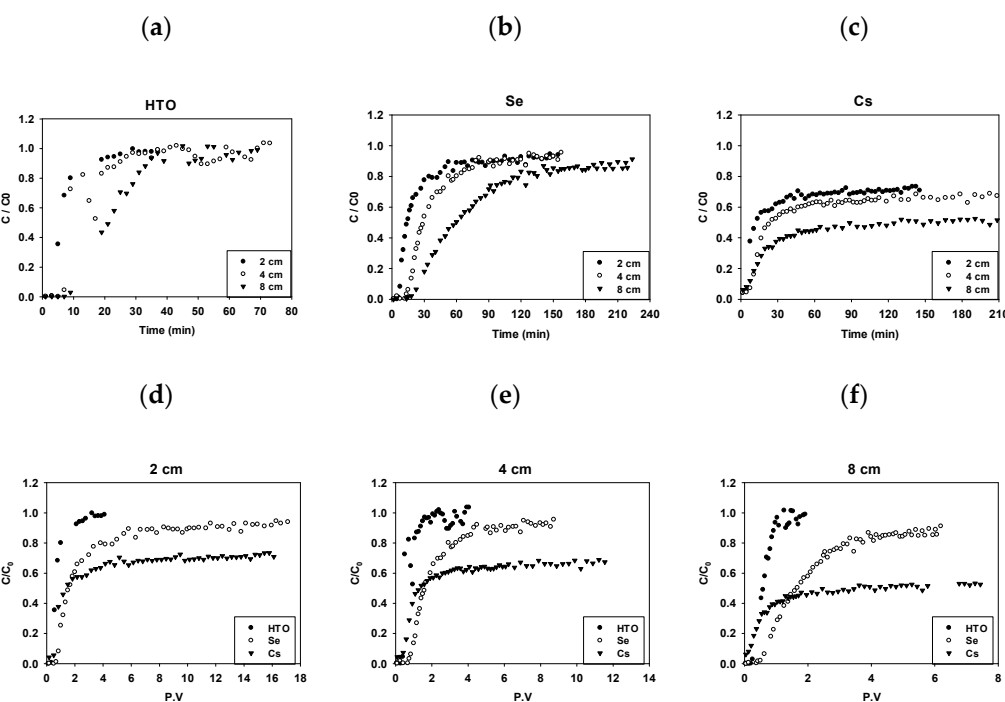

**Figure 3.** The experimental breakthroughs in different lengths. (**a**) HTO; (**b**) Se(IV); (**c**) Cs; (**d**) 2 cm; (**e**) 4 cm; (**f**) 8 cm.

*3.3. Simulation for Transport: HTO, Se, and Cs*

Among the models in STANMOD, the equilibrium and non-equilibrium models in CXTFIT were primarily used to evaluate the transport parameters based on the experimental data. Generally, the sum of squared residuals (*SSQ*) are the degree to which the experimental results (*Ce*) deviate from the predicted values (*Cp*), where the *RMSE* (root mean square error) is defined as:

$$RMSE = \sqrt{\frac{\sum (C_p - C_e)^2}{N}} = \sqrt{\frac{\sum (SSQ)^2}{N}} \tag{18}$$

The BTC data are shown in Figure 4, and the simulation results are summarized in Tables 2–4 for the equilibrium model and non-equilibrium regions and site models. The figure shows that there was no retardation phenomenon with the HTO retardation factor (*R* ≒ 1). In addition, the results for both of the non-equilibrium regions and the two transport model sites were better than that for the equilibrium model due to the lower residuals (*SSQ* or *RMSE*). However, Tables 3 and 4 show there was an obvious difference in the BTCs of Se(IV) and Cs, with a higher retardation factor (*R* >1). This indicated a kinetic sorption reaction of Se(IV) and Cs on the granite, which suggests suitable fitting and explanations using the different reaction transport models (see Figure 4d–i and Table 4).

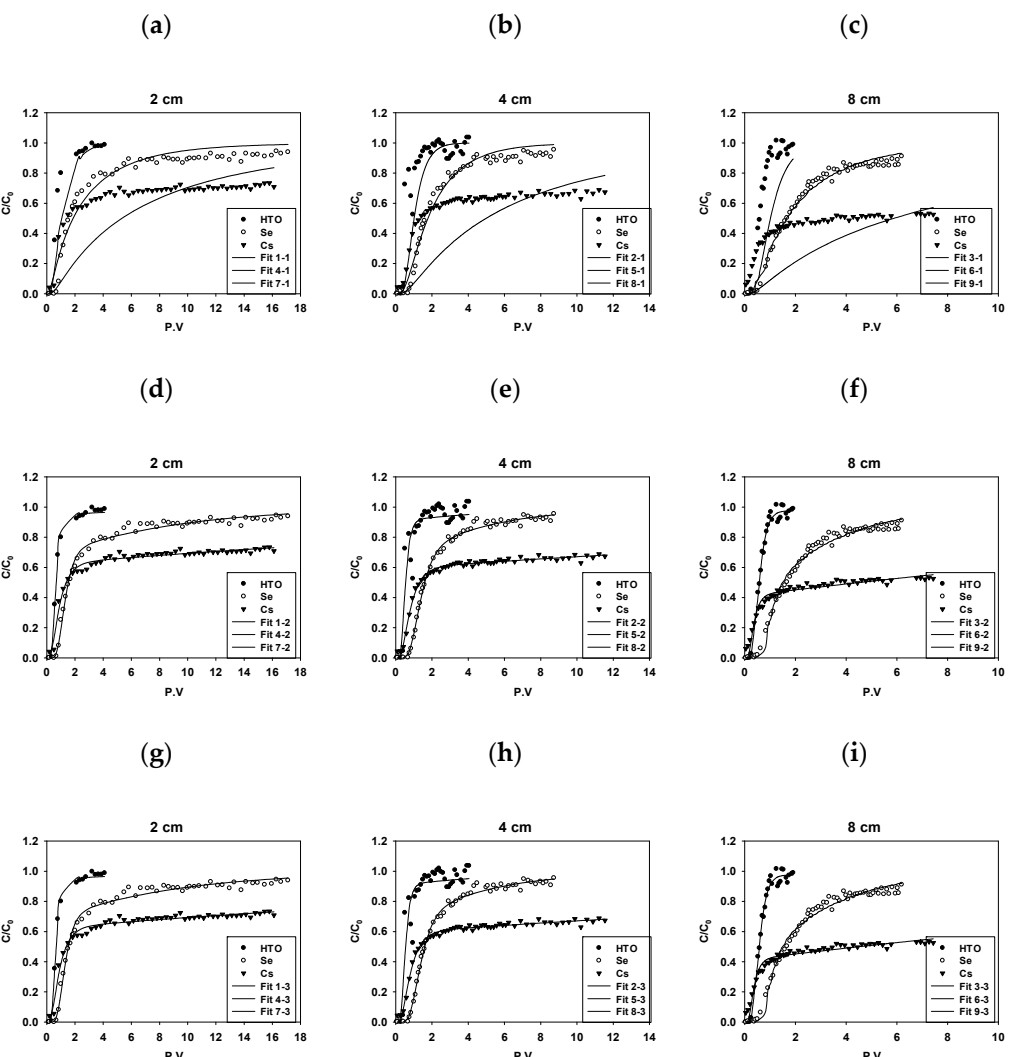

**Figure 4.** The experimental breakthroughs and curve-fittings for HTO, Se(IV), and Cs. (**a**) 2 cm-equilibrium model; (**b**) 4 cm-equilibrium model; (**c**) 8 cm-equilibrium model; (**d**) 2 cm-physical non-equilibrium model; (**e**) 4 cm-physical non-equilibrium model; (**f**) 8 cm-physical non-equilibrium model; (**g**) 2 cm-chemical non-equilibrium model; (**h**) 4 cm-chemical non-equilibrium model; (**i**) 8 cm-chemical non-equilibrium model.

**Table 2.** The fitting parameters for HTO, Se(IV), and Cs using the equilibrium model.

| RN | No. | $D$ (cm²/min) | $\theta$ (-) | $\alpha$ (cm) | $R$ | $SSQ$ | $RMSE$ |
|---|---|---|---|---|---|---|---|
| HTO | Fit 1-1 | 0.1158 | 0.46 | 0.53 | 1.00 | $2.76 \times 10^{-1}$ | $1.40 \times 10^{-1}$ |
| | Fit 2-1 | 0.1173 | 0.46 | 0.53 | 1.00 | $1.21 \times 10^{0}$ | $1.86 \times 10^{-1}$ |
| | Fit 3-1 | 0.2627 | 0.46 | 1.19 | 1.00 | $2.43 \times 10^{0}$ | $3.06 \times 10^{-1}$ |
| Se(IV) | Fit 4-1 | 0.4889 | 0.46 | 2.22 | 1.47 | $2.19 \times 10^{-1}$ | $6.68 \times 10^{-2}$ |
| | Fit 5-1 | 0.3296 | 0.46 | 1.50 | 1.67 | $1.35 \times 10^{-1}$ | $5.25 \times 10^{-2}$ |
| | Fit 6-1 | 1.2847 | 0.46 | 5.84 | 1.34 | $6.88 \times 10^{-2}$ | $3.39 \times 10^{-2}$ |
| Cs | Fit 7-1 | 0.5641 | 0.46 | 2.56 | 4.00 | $1.44 \times 10^{0}$ | $1.71 \times 10^{-1}$ |
| | Fit 8-1 | 0.8889 | 0.46 | 4.04 | 4.00 | $2.41 \times 10^{0}$ | $2.11 \times 10^{-1}$ |
| | Fit 9-1 | 3.0345 | 0.46 | 13.79 | 4.00 | $2.14 \times 10^{0}$ | $2.05 \times 10^{-1}$ |

**Table 3.** The fitting parameters for HTO, Se(IV), and Cs for the physical non-equilibrium model.

| RN | No. | $D$ (cm$^2$/min) | $\theta$ (-) | $\alpha$ (cm) | $R$ | $\beta$ | $\omega$ | *SSQ* | *RMSE* |
|---|---|---|---|---|---|---|---|---|---|
| HTO | Fit 1-2 | 0.0351 | 0.46 | 0.16 | 1.00 | 0.60 | 0.05 | $6.89 \times 10^{-3}$ | $2.22 \times 10^{-2}$ |
|  | Fit 2-2 | 0.0378 | 0.46 | 0.17 | 1.00 | 0.52 | 0.08 | $3.34 \times 10^{-1}$ | $9.77 \times 10^{-2}$ |
|  | Fit 3-2 | 0.0348 | 0.46 | 0.16 | 1.00 | 0.60 | 0.02 | $2.77 \times 10^{-2}$ | $3.26 \times 10^{-2}$ |
| Se(IV) | Fit 4-2 | 0.0351 | 0.46 | 0.16 | 3.53 | 0.37 | 0.32 | $3.47 \times 10^{-2}$ | $2.66 \times 10^{-2}$ |
|  | Fit 5-2 | 0.0378 | 0.46 | 0.17 | 2.44 | 0.58 | 0.29 | $2.36 \times 10^{-2}$ | $2.19 \times 10^{-2}$ |
|  | Fit 6-2 | 0.0348 | 0.46 | 0.16 | 2.21 | 0.45 | 0.66 | $6.44 \times 10^{-2}$ | $3.28 \times 10^{-2}$ |
| Cs | Fit 7-2 | 0.1175 | 0.46 | 0.54 | 14.30 | 0.07 | 0.40 | $2.41 \times 10^{-2}$ | $2.22 \times 10^{-2}$ |
|  | Fit 8-2 | 0.1011 | 0.46 | 0.47 | 15.50 | 0.06 | 0.45 | $1.34 \times 10^{-2}$ | $1.58 \times 10^{-2}$ |
|  | Fit 9-2 | 0.0820 | 0.46 | 0.38 | 13.00 | 0.03 | 0.84 | $3.01 \times 10^{-2}$ | $2.43 \times 10^{-2}$ |

**Table 4.** The fitting parameters for HTO, Se(IV), and Cs for the chemical non-equilibrium model.

| RN | No. | $D$ (cm$^2$/min) | $\theta$ (-) | $\alpha$ (cm) | $R$ | $\beta$ | $\omega$ | *SSQ* | *RMSE* |
|---|---|---|---|---|---|---|---|---|---|
| HTO | Fit 1-2 | 0.0584 | 0.28 | 0.16 | 1.00 | 0.60 | 0.05 | $6.89 \times 10^{-3}$ | $2.22 \times 10^{-2}$ |
|  | Fit 2-2 | 0.1454 | 0.24 | 0.35 | 1.00 | 0.52 | 0.08 | $3.34 \times 10^{-1}$ | $9.77 \times 10^{-2}$ |
|  | Fit 3-2 | 0.2319 | 0.28 | 0.64 | 1.00 | 0.60 | 0.02 | $2.77 \times 10^{-2}$ | $3.26 \times 10^{-2}$ |
| Se(IV) | Fit 4-2 | 0.0584 | 0.28 | 0.16 | 3.53 | 0.37 | 0.32 | $3.47 \times 10^{-2}$ | $2.66 \times 10^{-2}$ |
|  | Fit 5-2 | 0.1449 | 0.24 | 0.35 | 2.44 | 0.58 | 0.29 | $2.36 \times 10^{-2}$ | $2.19 \times 10^{-2}$ |
|  | Fit 6-2 | 0.2319 | 0.28 | 0.64 | 2.21 | 0.45 | 0.66 | $6.44 \times 10^{-2}$ | $3.28 \times 10^{-2}$ |
| Cs | Fit 7-2 | 0.1958 | 0.28 | 0.54 | 14.30 | 0.07 | 0.40 | $2.41 \times 10^{-2}$ | $2.22 \times 10^{-2}$ |
|  | Fit 8-2 | 0.3876 | 0.24 | 0.93 | 15.50 | 0.06 | 0.45 | $1.34 \times 10^{-2}$ | $1.58 \times 10^{-2}$ |
|  | Fit 9-2 | 0.5469 | 0.28 | 1.51 | 13.00 | 0.03 | 0.84 | $3.01 \times 10^{-2}$ | $2.43 \times 10^{-2}$ |

## 4. Discussion

The fine micro-fractures and pores within intact granite were analyzed using a micro-CT analysis. This is an important issue for radionuclide migration in a safety assessment due to the long disposal period. Furthermore, the ADE experimental results indicated that the dispersion coefficients ($D$) and retardation factor ($R$) (depending on the porosity and dominant minerals, such as biotite) are responsible for the sorption of Se(IV) and Cs.

In fact, it was found that the experimental data exhibited a poor fit using the equilibrium model, where all of the non-equilibrium models showed a better fitting result than the equilibrium model. On the other hand, the dispersion coefficients ($D$) tended to increase with column length because of higher dispersivity ($\alpha$), with a fixed flow rate or Darcy velocity. The increase in $\alpha$ with increases in $D$ may have been caused by an increase in the amount of immobile water, and hence by an increase in the diffusion length for the radionuclides from the mobile to immobile center [22].

Among all of the results, the retardation factor ($R$) of Cs showed a higher value than Se by the two-region and two-site transport models, which was in agreement with the findings of several other reports, because of more adsorption sites or frayed edge sites (FES) in biotite [29].

Scale-dependent dispersivity is very important for engineering designs in terms of providing appropriate dispersivity associated with the design scale. In this work, it should be noted that our laboratory study deals with relatively small column scales, with path lengths ranging from 2 to 8 cm and diameters of 5 cm, that are within an order of magnitude in centimeters. This small-scale heterogeneity may be observed in engineering studies due to the non-uniform packing and micro-structure of the materials. However, this is not very clear, and more experimental results should be provided indicating whether the relation of scale-dependent dispersivity can be extended to small-scale data such as the testing columns used in this study.

## 5. Conclusions

This study can be regarded as a long-term pilot study and the first attempt at conducting in-house transport experiments. In addition, these comparative experimental and modeling studies provided a way to extrapolate from laboratory scale to in situ conditions. Nevertheless, this study offers some interesting results about transport in granite, and, most importantly, it emphasizes the importance of analyzing both the breakthrough and leaching processes in granite. The conclusions drawn from the results of this study are as follows:

(1) In terms of the breakthrough process, there was a better fit of the experimental data using the non-equilibrium model as compared to the equilibrium model based on the lower residuals (*SSQ or RMSE*). Three sets of dispersion coefficients (*D*) were found when increasing the path lengths ranging from 2 to 8 cm in the breakthrough process (up-flooding or radiotracer introduction stage), suggesting higher dispersivity ($\alpha$) during the HTO switching process for Se(IV) and Cs. In the physical non-equilibrium models, the increase in $\alpha$ with increases in D may have been caused by an increase in the amount of immobile water, and hence by an increase in the diffusion length for the radionuclides from the mobile to the immobile center.

(2) The Cs transport was noted by the smaller values of $\beta$, where Cs had a higher retardation factor (*R*) than Se. Thus, a comparison of the micro-CT analysis images indicated that the micro-porosity in granite and dominate minerals, such as biotite, is mostly responsible for the sorption of Cs.

**Author Contributions:** Conceptualization, Y.-L.J., S.-C.T., C.-P.L. and N.-C.T.; methodology, C.-P.L., D.C. and N.-C.T.; software, C.-P.L. and N.-C.T.; validation, Y.S. (Yunfeng Shi), D.C., N.-C.T. and S.-C.T.; formal analysis, Y.S. (Yuzhen Sun), D.C., C.-P.L. and N.-C.T.; investigation, E.W., Z.W., Y.H. and Y.S. (Yunfeng Shi); resources, N.-C.T. and S.-C.T.; data curation, D.C., E.W., Z.W., Y.H. and Y.S. (Yunfeng Shi); writing—original draft preparation, C.-P.L. and N.-C.T.; writing—review and editing, C.-P.L. and N.-C.T.; visualization, Y.H. and Y.S. (Yunfeng Shi); supervision, C.-P.L. and N.-C.T.; project administration, N.-C.T. and S.-C.T.; funding acquisition, N.-C.T. and S.-C.T. All authors have read and agreed to the published version of the manuscript.

**Funding:** This project was majorly supported by the Doctor Initial Financial Project (No. 1410000434), East China University of Technology, and in part by the Ministry of Science and Technology (MOST, Taiwan R.O.C) and the Atomic Energy Council (AEC, Taiwan R.O.C) through a 2-year mutual fund program project under contract numbers 109-2622-E-007-022, 109-2623-E-007-006-NU, and 110-2623-E-007-004-NU.

**Acknowledgments:** The experimental and instrumental analysis of this study was supported by the Department of Nuclear Environmental Science, China Institute for Radiation Protection (CIRP), Taiyuan, China, Instrumentation Center at National Tsing Hua University, and the National Synchrotron Radiation Research Center (NSRRC) in Taiwan under contract number 2020-1-123-5.

**Conflicts of Interest:** The authors declare no conflict of interest.

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
